# A Review on the Out-Of-Autoclave Process for Composite Manufacturing

**Okunzuwa Austine Ekuase, Nafiza Anjum, Vincent Obiozo Eze and Okenwa I. Okoli ***

High-Performance Materials Institute, College of Engineering, Florida Agricultural and Mechanical University (FAMU)-Florida State University (FSU), Tallahassee, FL 32310, USA; okunzuwa1.ekuase@famu.edu (O.A.E.); na20@my.fsu.edu (N.A.); veze@fsu.edu (V.O.E.)

* Correspondence: okoli@eng.famu.fsu.edu

**Abstract:** Composite materials have gained increased usage due to their unique characteristic of a high-stiffness-to-weight ratio. High-performing composite materials are produced in the autoclave by applying elevated pressure and temperature. However, the process is characterized by numerous disadvantages, such as long cycle time, massive investment, costly tooling, and excessive energy consumption. As a result, composite manufacturers seek a cheap alternative to reduce cost and increase productivity. The out-of-autoclave (OoA) process manufactures composites by applying vacuum, pressure, and heat outside of the autoclave. This review discusses the common out-of-autoclave processes for various applications. The theoretical and practical merits and demerits are presented, and areas for future research are discussed.

**Keywords:** autoclave; out-of-autoclave; resin transfer molding; quickstep; composite

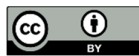

## 1. Introduction

The composite material is formed by combining a fiber reinforcement and a binding matrix [1]. The resulting material is lightweight but has high strength and stiffness [2]. Composite materials offer exceptional properties such as high thermal stability, flexural strength, damping property, corrosion resistance, impact resistance, and fire resistance [3]. Due to their exceptional properties, composite materials are useful in various industries such as aerospace [4], space exploration [5], construction [6], automobile [7], biomedical [8,9], sports [10], and marine [11]. Based on fiber types, composites are categorized as particle reinforced composites, discontinuous fiber-reinforced composites, and continuous fiber-reinforced composites. Composites made of fibrous reinforcements are stronger and stiffer than those made from particulates and are referred to as fiber-reinforced plastic (FRP). In FRP systems, the fiber acts as the load-carrying member, and the matrix binds the fibers together, protects the fibers from abrasion and the environment, and acts as a load transfer medium. Fibers commonly used in FRP systems are glass, carbon, aramid/Kevlar, and boron fibers. These fibers are combined with the polymer matrix in either a chopped or a continuous form. Based on the matrix used, composites are categorized into polymer matrix composites (PMC), metal matrix composites (MMC), ceramic matrix composites (CMC) [12], or hybrid composite materials [13]. PMC-wide usage can be attributed to its flexibility in fabricating complex and large shapes. Thermosetting or thermoplastic polymers are used as matrix components. Thermoplastic polymers can be subjected to repeated heating and cooling cycles. In contrast, a thermosetting polymer cannot be reversed after curing [14]. Commonly used thermoplastic polymers are polyethylene (PE), polypropylene (PP), and polyvinyl chloride (PVC), and examples of thermosetting polymers are epoxy, unsaturated polyester, polyimides, and bismaleimides [15]. Thermosetting polymers are often used to fabricate polymer-based composites because of their ease of processing.

Autoclave processing is typically used for fabricating fiber-reinforced plastic (FRP) composites for high structural applications [16]. Layers of fibers pre-impregnated with resin (known as prepreg) are stacked on a mold to form the desired component shape. The assembly is covered with different layers of bleeder and breather and then sealed with a vacuum bag. The bleeder helps absorb excess resin squeezed out from the laminate, and the breather creates a channel through which air and volatiles are ejected from the assembly [16]. The mold-laminate assembly is placed in the autoclave, a large, temperature, and pressure-controlled vessel. The bag is connected to the vacuum system, and a predetermined temperature and pressure (cure cycle) is applied to the laminate. The temperature initiates and sustains the chemical reaction to cure the resin. The pressure compacts the laminate to the desired fiber volume fraction and collapses any void present during curing. In addition, the pressure conforms the laminate to the tool surface. Several models have been developed to simulate autoclave curing for efficient processing [17]. For example, the cooling and reheating cure models were developed to prevent a thermal spike from an exotherm reaction, leading to partial degradation when curing a thick composite [18]. Furthermore, incorporating the smart cure monitoring model [19] has helped optimize the cure cycle in autoclave curing. An optimized autoclave curing is believed to reduce the cost of processing. Though the product of the autoclave process is a high-performance and reliable composite structure, many manufacturers are concerned with its numerous drawbacks. Some of its disadvantages are massive investments, excessive energy consumption, and costly tooling. Consequently, only the aerospace industries can conveniently afford the costs due to safety reasons. Most manufacturers are turning to other alternatives.

The out-of-autoclave (OoA) process manufactures composites by applying a vacuum, pressure, and heat outside the autoclave [20]. The OoA process uses lower pressure than the autoclave and cures composites in an oven or heat blankets. Hence, a special resin system is developed to evacuate voids efficiently [21]. Though the OoA process is more cost-effective than autoclave curing, the quality of composites manufactured by this process is still inferior to those processed in the autoclave [21]. This review discusses the common out-of-autoclave processes, their merits and demerits, and their applications. Future research direction on some OoA processes is also presented.

## 2. Prepreg

Prepregs are sheets of unidirectional fibers pre-impregnated with a partially (B-stage) cured resin matrix [22]. Prepregs are produced by placing fibers between two resin sheets, usually epoxies, and passing the fibers through rollers to achieve complete wet-out. In order to prevent premature curing, the wetted prepreg is wound up and stored in a refrigerator (typically at −18 °C) [22]. Prepregs are available in varieties of widths ranging from 3 inches to 72 inches, depending on the dimension of the machine used. Their thickness ranges from 0.01 mm to 0.8 mm, depending on the type of fiber form used. Common fiber prepregs include unidirectional tapes and woven and prepreg tows [23]. Different types of resin used for prepreg manufacturing are epoxies, phenolics, and cyanate esters [24]. Prepregs are very flexible, which permits them to be shaped to fit a complex mold. Furthermore, prepregs have sticky surfaces due to the partially cured resin, which facilitates the prepreg layers' stacking and prevents possible movement. Prepregs can be laid either by the manual lay-up process or by automation. A schematic of a typical prepreg is presented in Figure 1 [25].

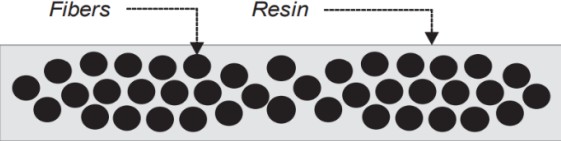

**Figure 1.** Prepreg [25]. Reprinted with permission from Elsevier.

### 3. Vacuum Bagging

The vacuum bagging process uses a flexible transparent film to enclose and compact wet laminates using atmospheric pressure. Figure 2 depicts the vacuum bagging process [26].

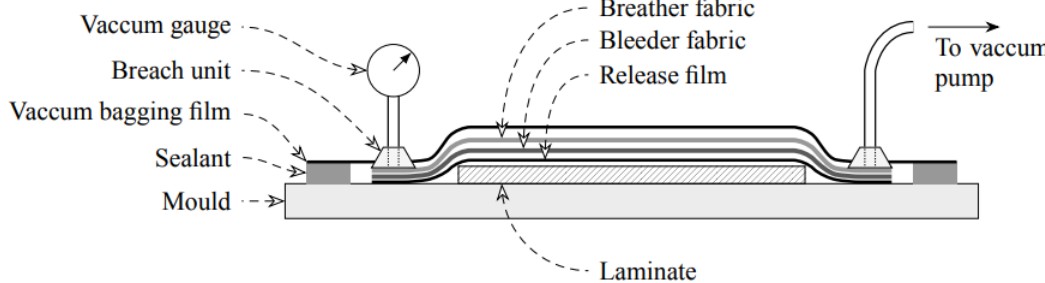

**Figure 2.** Vacuum Bagging Molding Process [26]. Reprinted with permission from Springer Nature.

This method uses a vacuum pump to extract the air inside the vacuum bag and then compresses the part under atmospheric pressure [27]. The resin is squeezed and sucked from the wet laminate into the bleeder (woven polyester fabric). Materials used in the vacuum bagging process are cheap, yet the parts fabricated with this process yield better mechanical properties than hand lay-up. Furthermore, the applied pressure is evenly distributed over the entire surface regardless of the quantity and type of material processed. The effect of evenly applied pressure is a thinner laminate with fewer voids [28]. Therefore, the process effectively controls excess resin in the laminate that increases the fiber volume fraction. Furthermore, it is a simple process that can use a variety of molds. However, some of the disadvantages of using this process are that with a bigger and more complex lay-up comes more support which increases labor. The process needs to be completed once started without having a break in between. Fiber volume fraction cannot be effectively calculated as other methods, mainly when over-bleeding occurs. The materials used for production in the vacuum bagging process are listed in Table 1. The vacuum bag technique can be used to fabricate yachts, primary structures such as decks, hulls, superstructures, bulkheads, and secondary structures such as partition panels and interior joint work [29]. The vacuum bagging process has shown considerable improvements in the mechanical properties of fabricated parts compared to hand lay-up processing. However, hand lay-up parts are inferior to parts manufactured by the vacuum infusion process. The vacuum infusion process will be considered later in this review.

There are two types of vacuum bagging processes, i.e., single vacuum bagging (SVB) and double vacuum bagging (DVB). In the SVB technique, prepregs are stacked between the tool and caul then a vacuum bag is used to seal the assembly. The assembly is installed in an air-forced oven, and heat is applied. At low temperature, usually in stage B, a vacuum is pulled on the inside of the vacuum bag to consolidate the composite. On the contrary, the DVB uses a double vacuum bag to efficiently evacuate volatiles produced during curing. With the normal setup of the SVB, the DVB installs an outer vacuum bag over the assembly with a perforated steel tool in between the two bags [30]. As shown in Figure 3, the perforated steel tool prevents the outer bag from collapsing into the inner bag when the vacuum is pulled. A full and partial vacuum is pulled on the outer and inner bags, respectively. The pressure differential results in a "ballooning effect" that aids the evacuation of entrapped air and volatiles. During the final curing stage of the laminates, the outer bag vacuum is purged to atmospheric pressure, and the inner bag is maximally vacuumed to consolidate laminates. Studies have demonstrated the escape of volatiles and entrapped air to be more efficient in the DVB technique than in the SVB technique.

**Table 1.** Function of Vacuum Bagging Components.

| Component | Function |
|---|---|
| Release Agent | Permits the release of the cured prepreg from the tool. |
| Peel Ply | A porous material that allows the passage of excess resin to flow through it during curing. It aids the removal of the bagging system from the parts after curing. An example of this material is a perforated Teflon sheet. |
| Release Film (separator) | A lightly porous material that permits the flow of air and volatiles only unto the breather; however, it restricts the further flow of resin. |
| Bleeder Fabric | Typically, a fiberglass mat that absorbs excess resin as it flows out during the molding process. |
| Breather Fabric | A highly porous material that allows the removal of air and volatiles from the composite assembly. Examples of breather materials are fiberglass, polyester felt, and cotton. |
| Vacuum Bag /Sealant Tape | A sticky polymeric tape placed around the entire assembly to provide an airtight seal vacuum bag. |

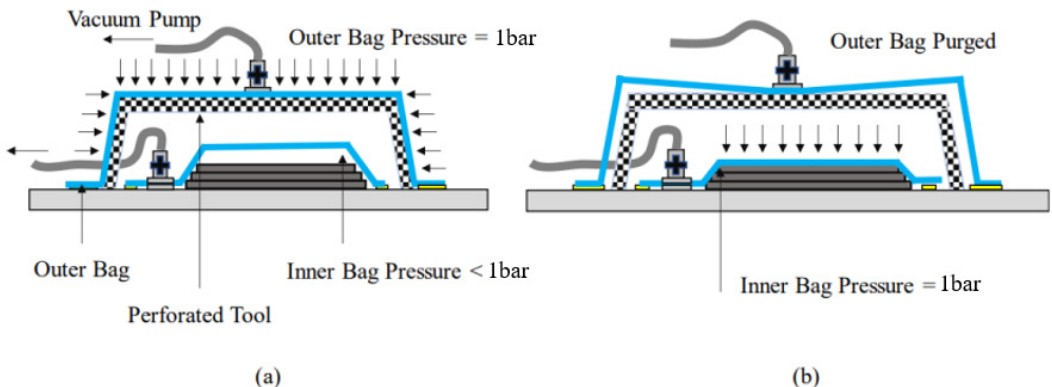

**Figure 3.** Schematics of a flexible double-vacuum-bagging setup with a perforated tool during (**a**) degassing phase and (**b**) compaction phase. Ref. [30] Reprinted with permission from Springer Nature.

Yasir et al. [30] compared the compatibility and performance of the DVB to the SVB methods for fabricating high-quality composites. The study used carbon prepreg with 58% fiber content. Their results show that laminates fabricated with DVB techniques had better performance than SVB fabricated laminates. The DVB laminates had surface porosity and through-thickness of 0.04% and 0.5%, respectively, while the SVB laminates had surface porosity and through-thickness of 0.1% and 0.7%, respectively. The DVB process provides a better pitting effect via the ballooning effect to allow the evacuation of volatiles during curing than the SVB process. The vacuum bag holds tightly to the laminates and leaves minimal space for entrapped air to be evacuated. Furthermore, it is observed that increasing laminate thickness increases the void content and hinders the evacuation of the entrapped air. In another study, Hou et al. [31] observed similar results assessing the void content of laminates fabricated by these two methods. Figure 4 illustrates the increased void content observed in laminates fabricated by SVB compared to those manufactured by DVB.

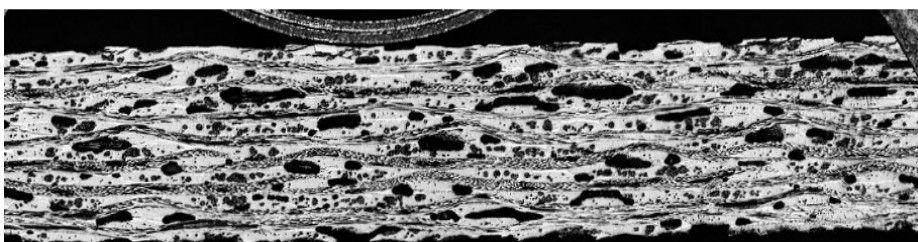
SVB molded laminate (20×)

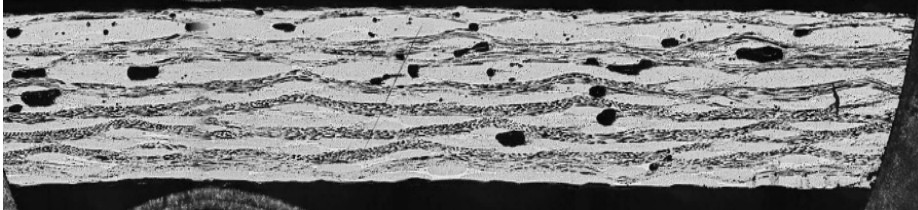
DVB molded laminate (20×)

**Figure 4.** SVB and DVB Micrograph. Reprinted with permission from [31].

Conversely, Khan et al. [32] did not observe significant results when they applied the DVB technique in the quickstep process to squeeze out excess resin during curing. Cytec cycom prepreg containing epoxy resin was used in the study. Results demonstrated that DVB techniques could remove small resin patches when the application of external pressure is controlled.

## 4. Vacuum Bag Only (VBO)/Oven Cure

Vacuum bag only (VBO) curing is an out-of-autoclave (OoA) technique for processing composite laminates. It is performed in a contemporary oven without external pressure, such as the autoclave, to consolidate the laminate. In the absence of elevated pressure, it is important to consider the OoA resin property, fiber bed architecture, and prepreg system. The OoA resin is a slow cure kinetics and low cure temperature matrix system. Figure 5 shows the manufacturing assembly of a vacuum bag only composite, with its consumables. The OoA prepregs are characterized by a partially impregnated microstructure that presents in-plane permeability, which permits air evacuation and aids the manufacturing of low-porosity parts without using autoclave pressure [33]. The partially impregnated microstructure includes dry spots and resin-rich regions. Low pressure of 0.1 MPa is available for consolidation during cure, and it is insufficient to prevent void formation [34]. Therefore, the entrapped air, moisture, and other volatiles in the laminate must be evacuated before the resin gels. As a result, the dry regions in the partially impregnated microstructure form an internal network that facilitates gas exit during the initial low-temperature stage of cure. At high temperatures, the dry areas are infiltrated by resin from the resin-rich region. Repecka and Boyd [35] reported that partially impregnated prepregs resulted in a void-free panel, while fully impregnated prepregs led to over 5% void content. An impregnation level of 60% has been found to produce void-free panels. However, Ridgard [36] highlighted that the degree of impregnation should be considered regarding resin viscosity, cure cycles, and laminate quality. Yang and Young [37] demonstrated that the degree of saturation of a VBO prepreg affects the mechanical properties of laminates. The laminates were made with epoxy resin. Fully impregnated carbon fiber and dry fibers were assembled as hybrid laminates, and different degrees of saturation were defined; over-saturation, saturation, and undersaturation. Laminates with over-saturation exhibited similar mechanical properties as those fabricated with the autoclave. For over-saturation to occur during VBO processing, conditions should favor the impregnation rate. Centea et al. [38] demonstrated that the thermal gradient of a partially impregnated prepreg affects the rate of impregnation and gas transport during

consolidation. The Cycom 5320-1 epoxy system was used for the investigation. Porosity distribution is shown to be influenced by the thermal gradient. Areas with hotter-than-average temperatures prevented air from evacuating the laminates. The study reported that resin flow, permeability, bubble transport, and temperature evolution affected air evacuation. Other parameters such as prepreg formats may affect laminates produced with VBO prepregs. Maguire et al. [39] investigated the importance of prepreg formats and the manufacturing method for VBO prepregs. Manually applying epoxy powder may lead to non-uniform powder distribution, which could produce better laminate uniformity. The study confirmed that epoxy powder prevents an exotherm reaction in thick composites. However, the temperature cycle and latency of the epoxy powder need to be optimized for the best results. How the epoxy powder propagates heat within the VBO prepreg was unclear to the authors; hence further investigation is required. In another study, Edward et al. [40] designed a unidirectional semi-prepreg that improved the robustness of VBO processing. A toughened epoxy resin was used. The semi-prepreg was customized to discontinue resin distribution. As a result, through-thickness permeability was improved, which facilitated gas evacuation. Laminates produced by the semi-prepreg had fewer defects than those produced by conventional VBO prepregs. The resin feature morphology was observed to be critical in defect formation.

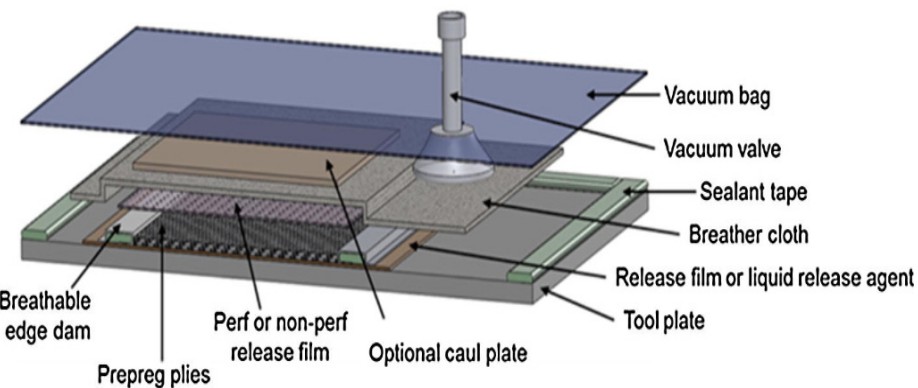

**Figure 5.** Manufacturing assembling of VBO cure [21]. Reprinted with permission from Elsevier.

Hu et al. [41] observed that vacuum quality (between 80–100%) does not affect the final porosity of VBO laminates. However, with 80% vacuum, the bubbles were observed to expand at the intermediate cure. Water molecules were reported to be responsible for this expansion, and they dissolved into the resin before the cure cycle ended. However, increased moisture content increases porosity and prevents the evacuation of air. Porosity between inter-ply is determined by entrapped air that is not evacuated prior to consolidation. The application of an appropriate cure cycle was shown to mitigate defect formation in VBO processing. Park et al. [42] optimized the cure cycle for VBO prepreg to minimize defect formations such as surface porosity and void content. The authors used a carbon-fiber-toughened epoxy prepreg system for the investigation. They observed that an isothermal dwell at 130 °C for 30 min would cause adequate infiltration of resin into the dry areas of the prepreg. Infiltration occurs when resin viscosity is reduced, and fibers are thoroughly wet, resulting in improved fiber/matrix interfacial bonding. Laminates produced with an optimized cure cycle have similar mechanical properties to autoclave cure laminates. Conversely, Yoozbashizadch et al. [43] held the isothermal cure and post-cure temperatures for eight hours at 181 °C and 211 °C, respectively, of a carbon fiber BMI prepreg system and attained an optimal ILSS value for a VBO prepreg. However, using an eight-hour cycle each for a cure and a post-cure will amount to a long cure cycle, increasing the cost of manufacturing. Therefore, a shortened cure cycle would be economically competitive for VBO processing. Hyun et al. [44] reduced the cure cycle duration for VBO prepreg processing and still maintained improved part quality. Cycom 5320-1 epoxy

system was used for the experiments. The study showed that applying an isothermal dwell at 60 °C for two hours will produce parts with no wrinkles and less porosity than a 16 h RT vacuum hold [45]. Entrapped air is efficiently removed at a moderate temperature when evacuation channels are not yet collapsed. In addition, they observed that eliminating the intermediate dwell at 121 °C for two hours shortened the cure cycle yet improved the mechanical properties. The question then becomes if a shortened cure cycle will improve mechanical properties for variously shaped parts. In this light, Mujahid et al. [46] investigated how different curing profiles interact with other VBO processing parameters, such as bagging techniques and laminate structures, to minimize defects. A 58% fiber-rich OoA epoxy prepreg was used in the study. The modified single vacuum-bag-only technique exhibited fewer defects and thickness variations in the fabricated part than other bagging methods. The laminates fabricated with the convex mold had better quality than the concave mold. However, concave parts can be improved by increasing the local curvature angles and corner radii of the mold.

Several studies have demonstrated the traditional cure kinetic behavior of OoA resins. The resin is cured with an initial increase rate, followed by a decrease due to the diffusion-based cure mechanism. Gelation occurs, and then vitrification of resin takes place during complete polymerization. The OoA resin is more reactive and more viscous than the autoclave resin for a cure cycle of 121 °C dwell time [47]. The OoA resin systems undergo a freestanding post-cure at 177 °C.

The OoA prepreg is consolidated by applying a vacuum at room temperature to evacuate the vacuum bag, compact the part and push out voids in the laminate toward the vacuum source, as illustrated in Figure 6. As a result, the fiber volume fraction increases, and the in-plane permeability decreases in the prepreg. During this time, the resin flow is limited due to the high viscosity of the matrix. When the part's temperature is increased, the resin viscosity decreases such that there is a progressive infiltration of the fiber bed by the resin. Resin flows into the dry fiber tows and saturates the interlaminar spaces. Impregnation for OoA prepregs is usually completed at the end of the first temperature ramp. Based on the impregnation rate, the dry evacuated channels are saturated once the dwell temperature is reached. In the last stage of consolidation, the resin undergoes gelation and vitrification, and then the cure is complete.

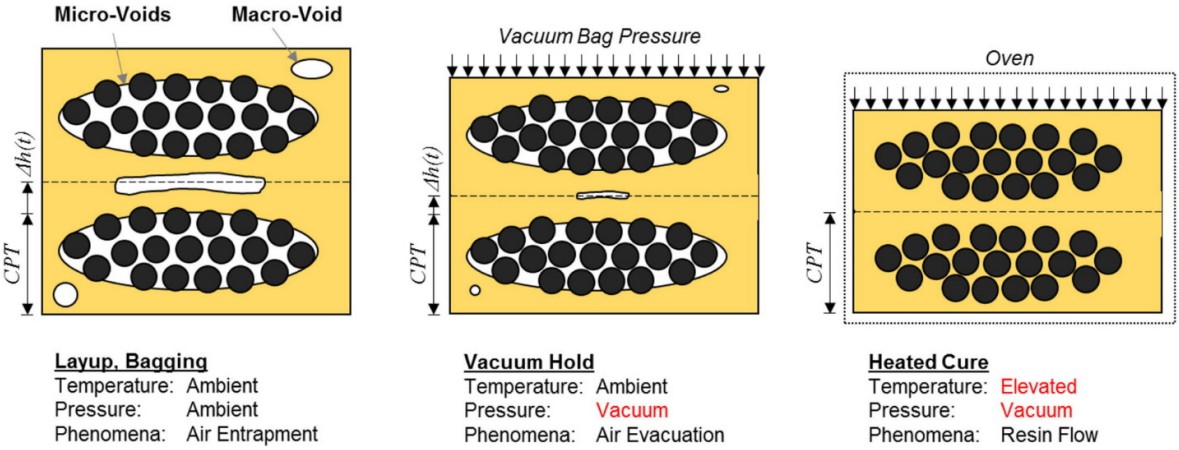

**Figure 6.** Consolidation Process of OoA Prepreg [21]. Reprinted with permission from Elsevier.

Porosity is formed due to incomplete resin flow into dry areas of the reinforcement. It is commonly caused by increased resin viscosity due to exposure to ambient conditions and low-temperature cure cycles, resulting in an insufficient flow. Three types of voids are identified in the VBO prepreg; these are spherical voids in resin-rich regions, interlaminar voids, and voids within fiber bundles. However, a "super-ambient" cure cycle that dwells at 50 °C to 60 °C for four hours has been found to be effective in achieving

low-level porosity [48]. Dong et al. [49] minimized the cure cycle time of the VBO process and studied how heating rate, initial cure temperature, dwelling time, and post-cure time affect the final quality of the composite. The authors developed an independent OoA epoxy resin for the study. They observed that a faster heating rate of 5 °C/min results in lower resin viscosity and saves 35% of cure time than a heating rate of 1.5 °C/min. The optimized dwell time of one hour at 120 °C allows enough resin flow time for fibers to be adequately infiltrated by low viscosity resin. Hence, improving fiber/matrix interfacial bonding and leading to enhanced mechanical properties. Furthermore, the authors concluded that an optimal post-cure could be carried out at 180 °C for two hours. Below or above this time duration would reduce mechanical properties or result in extra manufacturing costs. Some of the OoA prepregs are listed in Table 2.

**Table 2.** Current-Generation Aerospace-Grade OoA/VBO Prepreg Resin Systems [21]. Reprinted with permission from Elsevier

| Manufacturer | Resin Family | Resin type | Description |
|---|---|---|---|
| ACG (now Cytec) | MTM44-1 | Epoxy | Medium temperature molding (MTM) toughened epoxy. Qualified by Airbus® for secondary and tertiary structure. |
| | MTM45-1 | Epoxy | Lower temperature cure system optimized for compression performance. |
| Cytec | Cycom 5320 | Epoxy | Toughened epoxy designed for primary structure application. |
| Hexcel | Hexply M56 | Epoxy | High-performance VBO epoxy system. |
| Tencate | BT250E | Epoxy | Standard VBO system used in circus aircraft and unmanned vehicles. Variations for fatigue and fracture resistance for helicopter rotor blades. |
| | TC250 | Epoxy | Second generation VBO system with increased toughness and higher service temperature. |
| | TC275 | Epoxy | Third generation system with greater inspectability, resistance to hot/wet conditioning, and curable at 135 °C. |
| | TC350-1 | Epoxy | Third generation system with increased out-life (45 ± days), high toughness, and ability to cure at 135 °C with 177 °C required post-cure. |
| Henkel | Loctite BZ | Benzoxazine | VBO prepreg based on a blended epoxy-benzoxazine resin formulation. |
| Toray | 2510 | Epoxy | Formulated to meet the requirements of general aviation primary structure. |

## 5. Resin Transfer Molding (RTM)

The RTM process involves using a closed mold to fabricate a composite part. Figure 7 presents the various steps in the RTM process. Fiber preform is cut according to the mold shape and placed in a closed mold cavity [50]. A low-viscosity thermoset resin is injected through the injection port into the mold cavity, usually with a 3.5–7 bar pressure. The injected resin impregnates the preform evacuating entrapped air bubbles until complete wetting is reached. Once the resin starts exiting from the vent ports, the resin injection is stopped, and vent ports are closed. The resin is allowed to cure by heating the mold or the initial addition of inhibitors to the resin system. After the resin is cured, the mold is

opened, and the part is de-molded. Some variants of the RTM process are VIPR, FASTRAC, light RTM (LRTM), structural reaction injection molding (S-RIM), and co-injection resin transfer molding. Some advantages of RTM are that the process can produce parts with close dimensional tolerance and an improved surface finish. Parts made by RTM have a high-volume fraction of about 60–70%. RTM can manufacture complex-shaped composite parts. Consistent reproducibility of composite parts can be achieved using the RTM process. Due to high resin pressure and faster mold opening and closing, a fast-manufacturing cycle is reached, further improved by process control. Some drawbacks of the RTM process are the limited size of parts that can be manufactured. Fiber wash can occur due to high resin pressure and loose fiber compaction. Furthermore, improper location of injection gates and vents can lead to a macro void in the composite [51].

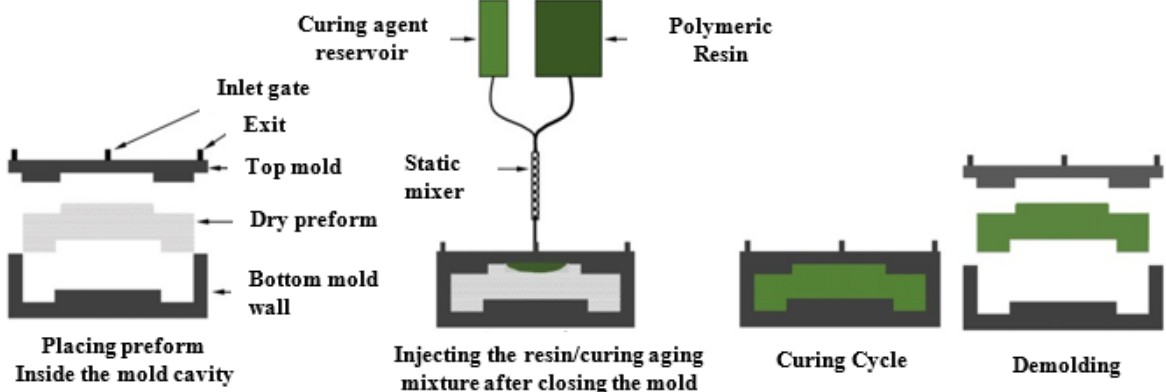

**Figure 7.** Schematics of Resin Transfer Molding (RTM) [51]. Reprinted with permission from Elsevier.

In RTM, some manufacturing issues include race tracks, deformation of fiber structure, and macro void formation. The region close to the fiber walls has higher permeability; therefore, the resin race along the path of high permeability leading to race tracking [52]. Race tracking occurs along the mold edges, along the ribs of bends [53], and on the lines of injection gates. In RTM, when the fiber is draped over the mold surface, fiber orientation in the preform is altered; hence permeability to resin flow is changed. The extent of deformation and permeability change is defined by fabric type and the radius of the mold curvature. There are four fiber deformation mechanisms: inter-fiber (intra-ply) shear, inter-fiber slip, fiber buckling, and fiber extension [54]. Deformation can cause fiber misalignment that leads to mechanical properties degradation. Vallon et al. [55] observed a 9% and 22% reduction in stiffness and strength, respectively, for a fiber misalignment of 5°. Li et al. [56] also found a 4% reduction in elastic modulus for every 1° of fiber misalignment up to 20°. Hsiao and Daniel [57] observed a 50% and 70% reduction in compression stiffness and strength, respectively, in S-glass/epoxy composites for a wrinkle value of 0.2. Kugler and Moon [58] observed a 20% reduction in a carbon/polysulfone laminate wrinkling with a lowering cooling rate from 20 °C/min to 2 °C/min.

The kinematic drape model [59] is used to predict fiber shear deformation and its effect on fiber volume fractions. Commercial software such as FiberSIM® and laminate-Tool [60] provide the draping angle for the lay-up; this helps adjust the flow pattern and predict the time to fill due to the draping fabric. Macro void occurs when resin flow arrives at the vents before the preforms impregnation is completed. Macro void is likely to happen when air is present in the preform and resin pressure is not high enough to collapse the void. Macro voids can be eliminated by bleeding the resin via the vent by allowing sufficient time or by using process control monitoring. RTM process parameters such as resin characteristics, fiber preform, resin preheated temperature, injection pressure, gating

method, mold geometry, mold temperature, and vacuum assistance have been studied and identified to influence the quality of parts molded using RTM. Combining the peripheral gating system and the vacuum resin transfer method results in a shorter injection time, minimizes void content, and increases fiber volume fraction and flexural strength compared to combining the positive pressure injection method and the peripheral gating system [61]. The radial gating arrangement decreases permeability which resists the resin flow resulting in a differential macroflow and microflow, leading to void formation [62]. Cevdet et al. [63] found 2 atm to be the optimum injection pressure and resulting in the highest flexural strength and modulus for injecting resin in the RTM technique. However, increasing injection pressure beyond 2 atm decreased the mechanical properties of the panels. The decreased mechanical properties were attributed to the difference in microflow and macroflow, leading to non-uniform resin flow and void formation. Increased injection pressure leads to fiber misorientation, reduced fiber volume fraction, and mold filling time. Moderate resin flow rate, low binder concentration on both sides of the fabric, and vacuum assistance have been reported as the most favorable parameters in fabricating composites for both injection and compression high-pressure RTM [64]. Compression pressure and increased temperature effectively reduce the void content [65].

## 6. Vacuum Assisted Resin Transfer Molding (VARTM)

In the VARTM method, the reinforcement is placed on a one-sided mold and sealed with a vacuum bag to form a closed mold. A vacuum is applied at the vent, which drives the resin under atmospheric pressure to impregnate the reinforcement while evacuating the air bubbles and compacting the fiber preform (Figure 8). The resin flows through the porous preform and arrives at the vent. The injection is closed, but the vacuum is maintained until the part is completely cured and de-molded. The VARTM process is used to produce large composite parts at a low cost with a low production volume [66]. This process is widely used in the energy, aerospace, marine, defense, and infrastructure building industries [67]. Variations of VARTM have been invented to cater to the manufacturing of complex parts with better quality at a reduced cost. The VARTM process has some advantages: flexibility of mold tooling and selection of mold materials [68], resin and catalyst can be stored separately and mixed before infusion, low emission of volatile organic compound (VOC), and visible inspection of the process to identify and manage dry spot occurrence [69]. However, some drawbacks of this process are that consumables such as sealing tape, peel-ply, and vacuum bags may not be reusable. The low resin injection pressure can limit void compressibility resulting in high void content and low fiber volume fraction. The process may be susceptible to high chances of air leakage, depending on the operator's skill level [70].

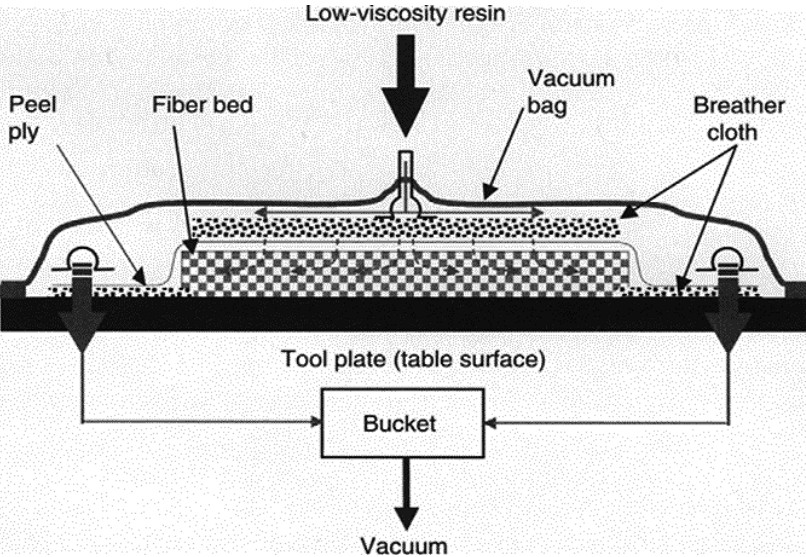

**Figure 8.** Schematics of VARTM [71]. Reprinted with permission from Elsevier.

The basics of the VARTM process are the resin flow phenomenon, fiber preform compaction, and resin viscosity [71]. The resin flow in the VARTM process is treated as flow-through anisotropic porous media and can be modeled by Darcy's law, represented as Equation (1).

$$u = -\frac{K}{\mu}\nabla P \tag{1}$$

where $u$ is the Darcy velocity, $\mu$ is the dynamic viscosity of the fluid, $P$ is the fluid pressure, and $K$ is the permeability of the stationary porous media. Equation (1) quantifies the relation between the Darcy velocity and resin-saturated pressure in a porous medium. Liquid resin in a solid porous medium requires the mass flow continuity for incompressible fluid and the solid, which is stated as

$$\nabla.\overline{U}_D = 0 \tag{2}$$

Equations (1) and (2) are combined to create a resin-saturated porous medium, which is stated as

$$0 = \nabla.\left(\frac{k}{\mu}.\Delta P\right) \tag{3}$$

When a boundary condition is assigned to the resin-filled porous domain, the pressure distribution inside the resin-filled domain can be solved by Equation (3). Equation (1) can then be used to solve the Darcy velocity distribution in the resin-saturated porous medium domain.

In the VARTM process, predicting the mold filling process is useful for determining major processing parameters and design windows. Hsiao et al. [72] proposed a two-dimensional analytical solution that uses dimensionless analysis to divide the resin-saturated porous medium domain into the saturation and flow front regions. The authors observed that flow front velocity in the VARTM process decreased significantly as the saturated region length increased. However, this effect can be minimized using a thicker flow distribution medium with higher permeability and preforms with higher permeability. The flow process is designed by determining the location of the injection gates and vacuum ports, sizes, and locations of the distribution lines, the number of layers, types and locations of flow distribution mediums, and finally, the timing to open and close the gates

and vents. In VARTM, the preform compaction pressure is the difference between the atmospheric and vacuum pressure (local pressure) inside the fiber preform. The fiber preform compaction affects the resin infusion process due to the change in preform permeability, thickness, and porosity. However, the influence on the thickness of the final part may not be too obvious if adequate relaxation time is not allowed for compaction pressure to be evenly distributed in the vacuum bag after the injection gate is closed. Longer duration between the resin filling and the resin gelation point will allow a complete relaxation process, thus improving thickness uniformity in the VARTM part. Bekir et al. [73] investigated the effect of compaction pressure and resin flow on part thickness variation in the vacuum infusion process. A polyester polipol matrix system was used. The study demonstrates that the preform is compacted effectively due to the lubrication effect with the initial resin injection. However, as the flow front advances, the compaction pressure is reduced, increasing laminate thickness. The authors, therefore, reported the duration of initial vacuuming and gelation, resin pressure, compaction pressure, and resin shrinkage ratio to determine the part thickness. Yacinkaya et al. [74] studied the effect of compaction pressure (CP) and infusion pressure (IP) in the fabrication of laminate panels using the pressurized infusion (PI) technique illustrated in Figure 9. The authors used an epoxy matrix in the study and observed that an increase in compaction pressure (CP) reduces the porosity and permeability of the fiber preform. While increasing the infusion pressure (IP) increase the porosity and permeability. Both CP and IP increase the fiber volume fraction and interlaminar shear strength and decrease the void content. Increasing the CP reduces laminate thickness and increases the fiber volume significantly. Increasing the CP and IP together reduces the void content.

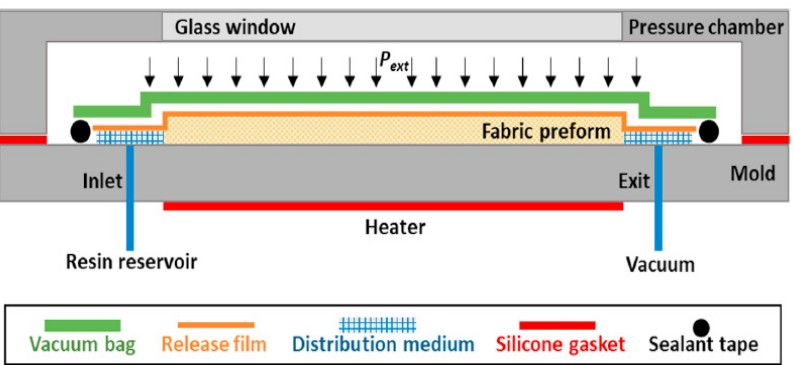

**Figure 9.** Pressurized VARTM Setup [74]. Reprinted with permission from Elsevier.

Furthermore, Yacinkaya et al. [75] investigated the synergetic effect of the external pressure coupled with resin flushing to enhance the quality of fabricated composites in a heated-VARTM system. A glass/epoxy system was used in the study. Applying external pressure alone decreases the average void content from 24% to 1.4%. However, coupling the external pressure with resin flushing further reduces the void content to 0.86%. Apart from pressurized air, permanent magnets have been used to generate compaction pressure. Maya et al. [76] used a magnet-assisted composite manufacturing technique to generate consolidation pressure during cure. The authors observed increased flexural properties, fiber volume fraction, and low void content. Kedari et al. [77] studied the effect of vacuum pressure, inlet pressure, and mold temperature on the VARTM process. Mold temperature, inlet pressure, and vent vacuum pressure were varied, fabricating different polyester/E-glass fiber composite system samples. Experimental results suggest that increased mold temperature and vacuum pressure at the vent increase the fiber volume fraction in a VARTM system. However, void content can considerably increase if the inlet pressure of 1.013 bar is not appropriately modified when mold temperature is increased. The authors identified vent pressure, inlet pressure, and mold temperature as the major

factors influencing void formation. Kedari et al. [77] reported the possibility of minimizing the microvoid content and increasing the fiber volume fraction by controlling mold temperature and resin inlet pressure. Controlling laminate thickness is critical in the mold and post-mold filling stages. The post-filling compaction relaxation process is influenced by the preform and fiber system, resin viscosity and cure kinetics, mold temperature, and the type and arrangement of the flow distribution network. Extending the resin processing window and keeping injection gates closed during the post-filling stage can improve uniformity in part thickness. Furthermore, increasing the mold filling speed and using resins with longer gel times can help control part thickness. Yacinkaya et al. [78] compared the compaction pressure between the VARTM and RTM methods. Increasing compaction pressure was found to decrease part thickness. However, changes in resin viscosity affected how compaction pressure changed. Resin viscosity, usually less than 1000 cP, is critical for mold filling, fiber preform compaction, and curing in the VARTM process. Mold temperature can be effectively used to control the resin viscosity of a high-performance resin system. Mold temperature selection is influenced by the type of mold material, resin gel time control, resin viscosity, resin curing management, flow medium distribution material, and peel-ply [79]. The mold is heated during the mold filling process to reduce resin viscosity and increase the processing window of the resin.

The challenges of the VARTM process are air entrapment, thickness and fiber volume fraction uniformity, curing and thermal management, and spring-in. Low vacuum can lead to the entrapment of air inside the composite part when resin flow fails to displace air; hence dry spot formation occurs. Optimizing the mold filling design can solve the problem of air entrapment. Dry spots also occur due to the slow mold filling process. Increasing the number of flow distribution layers, injection ports, and vents can be used to mitigate this problem. Hsiao et al. [80] designed and proposed a distribution media layout to manipulate the flow front, reducing race track. Reducing the resin curing rate or increasing the mold temperature can also be helpful. Leakage in the vacuum bag, resin supply line and sealing tapes can cause dry spots. Incompatible dual scale flow behavior between resin flow in the fiber tow and between the fiber tow can result in microvoid, another form of air entrapment. VARTM curing can lead to a potential thermal degradation in the thick parts. To avoid thermal spiking in a VARTM, White and Kim [81] proposed multi-stage curing (MSC). The MSC technique prevents thermal degradation, affecting the interlaminar fracture toughness and the interlaminar shear strength of composite parts.

The MSC process uses a combination of various "stage VARTM processes" that can only cure manageable layers of composites at a time to fabricate the thick laminate. The final composite thickness is different due to resin thermal contraction and volumetric shrinkage. Residual stress or strain occurs in the laminate due to resin cross-linking shrinkage and fiber and matrix mismatched thermal contraction. This residual stress–strain can cause dimensional problems called spring-in. Spring-in is the inward bending of the curve-shaped laminate part caused by curing. Laminates shrink in the thickness direction due to thermal contraction and cross-linking shrinkage, depending on the matrix. The fiber maintains the in-plane dimensions during the curing process. During the curing of the curve-shaped laminate part, the non-isotropic dimensional changes in the in-plane and thickness direction will result in the further inward bend of the laminate after de-molding.

VARTM is used to manufacture bio-based composites consisting of cellulose fiber mats and oil-based resins. The VARTM process has been identified as a promising method for manufacturing nano-enhanced FRP. The VARTM process reveals a unique, through-thickness, mold-filling flow pattern. The through-thickness flow reduces the nano-modified resin's traveling distance. Furthermore, the change of nanoparticles is filled by the fiber preform. Fan et al. [82] reported a 0.5 wt. % of MWCNT composite made with the VARTM process had CNTs aligned in the through-thickness direction, which improves the laminates' mechanical properties.

## 7. Quickstep Curing

In the quickstep process, prepregs are stacked up in a one-sided mold to form a laminate and sealed with a vacuum bag. The laminate-mold assembly, shown in Figure 10, is placed inside a pressure chamber supported by two flexible membranes. A heat transfer fluid (HTF) system controls the laminates' temperature and regulates resin viscosity by circulating the HTF through the pressure chamber [83]. The HTF, with high heat capacity and thermal conductivity, maintains a rapid heating and cooling rate as low pressure of 10 kPa is applied [84]. To further increase the laminate compaction and reduce voids, an alternating pressure is applied to the HTF. The quickstep process reduces the curing cycle, capital, tooling, and operational costs. Furthermore, the quickstep can fabricate medium composite parts of high quality. Nevertheless, the fact that the heat transfer system solely depends on fluid can be a disadvantage. The quickstep process may be restricted to medium complexity parts because of the low applied pressure. The flexible membrane has a limited life span [85]. Numerous studies have demonstrated the use of quickstep curing to fabricate laminate panels that are comparable to panels produced by the autoclave [86] and better than those produced by hot press and oven cured [87]. Enhanced composite properties made by quickstep are attributed to fiber bridging, consistent curing, and improved fiber/matrix adhesion in quickstep techniques [87].

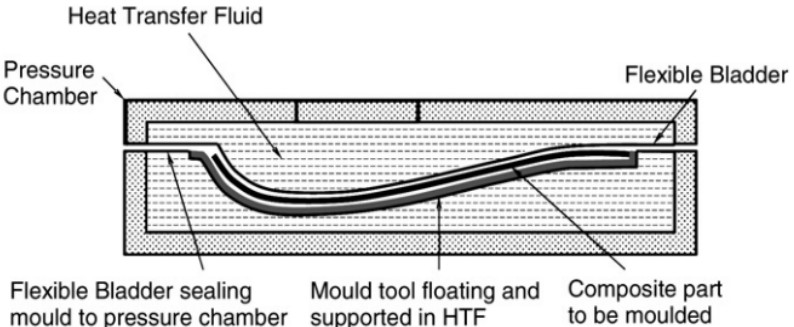

**Figure 10.** Schematics of the Quickstep Process [87]. Reprinted with permission from Elsevier.

The high heating rate in quickstep processing can quickly reduce the viscosity of the resin, as illustrated in Figure 11. As the heating rate increases, the minimum viscosity and the time it takes for the resin to reduce to the minimum viscosity decreases. Reduced resin viscosity improves fiber wetting, resulting in fiber/matrix adhesion and thus mechanical property enhancement. Davies et al. [88] optimized quickstep curing by designing three different cure cycles and comparing them with autoclave curing. They used a 10 °C/min heating rate to cure the quickstep composites and 2 °C/min for autoclave samples. Hexply 6376 prepreg containing epoxy was used in the study. It was observed that the quickstep samples had similar mechanical properties to the autoclave samples, as illustrated in Tables 3 and 4. Isothermal hold has been found to be critical in optimizing the quickstep cure cycle. Khan et al. [89] showed that the duration of isothermal hold, when the resin viscosity is at a minimum, is critical to enhancing mechanical properties. Three different cure cycles were designed and analyzed. A Cycom 977 prepreg containing epoxy was used. From Table 3, the $QS_{AD}$ cure cycle was manipulated for a prolonged processing window at resin minimum viscosity. The $QS_{ID}$ had an intermediate dwell, and the $QS_{Direct}$ had no isothermal dwell. Results reveal that the $QS_{AD}$ samples had better mechanical properties due to improved consolidation because of extended resin flow time. Furthermore, increasing the cure temperature and extending the dwell time increased glass transition temperature and interlaminar shear strength properties [90].

**Table 3.** Quickstep Different Curing Cycles.

| Cure Cycle | 1st RT °C/min | 1st D/T °C & min | 1st CT °C/min | 2nd RT °C/min | 2nd D/T °C & min | 3rd RT °C/min | 3rd D/T °C & min | Ref. |
|---|---|---|---|---|---|---|---|---|
| Q1 | 17 | 132/45 | | | | | | |
| Q2 | 17 | 132/60 | | | | | | [90] |
| Q3 | 17 | 140/45 | | | | | | |
| Q4 | 17 | 140/60 | | | | | | |
| Q1 | 10 | 175/0.02 | | 5 | 125/120 | | | |
| Q2 | 10 | 135/35 | | 5 | 175/60 | | | [84] |
| Q3 | 8 | 110/30 | | 7 | 180/60 | | | |
| A1 | 55 | 135/30 | | 20 | 175/60 | | | |
| QS$_{Direct}$ | 10–12 | 180/180 | | | | | | |
| QS$_{ID}$ | 10–12 | 130/60 | | 10–12 | 180/120 | | | [89] |
| QS$_{AD}$ | 10–12 | 175/10 | 10–12 | | 130/35, 45, 60, 75 | 10–12 | 180/120 | |
| Autoclave | 2 | 130/60 | | 2 | 180/120 | | | |
| QSspike | 10 | 175/0 | 8 | | 130/120 | 10 | 175/120 | |
| QSdwell | 10 | 130/20 | | 10 | 175/120 | | | [88] |
| QSstraight | 10 | 175/120 | | | | | | |
| Autoclave | 2 | 175/120 | | | | | | |

**Table 4.** Mechanical Properties of Different Quickstep Cure Cycles (in Table 3).

| Cure Cycle | Fiber Vol. Fraction | Void Content | ILSS | Flexural Strength | Glass Temperature | Ref. |
|---|---|---|---|---|---|---|
| Q1 | | | 61.6 | | 150 | [90] |
| Q2 | | | | | 153 | |
| Q3 | | | | | 153 | |
| Q4 | | | | | 157 | |
| Q1 | | | | | 208 | [84] |
| Q2 | | | | | 207 | |
| Q3 | | | | | 207 | |
| A1 | | | | | 200 | |
| QS$_{Direct}$ | 54.6 | 6.1 | 54 | 1019 | | [89] |
| QS$_{ID}$ | 57 | 4.8 | 65 | 1129 | | |
| QS$_{AD}$ (60 min) | 61 | 1.7 | 84 | 1258 | | |
| Autoclave | 63 | 0.8 | 81 | 1389 | | |
| QSspike | 60 | 1.8 | 115 | 1755 | | [88] |
| QSdwell | 55 | 8.1 | 84 | 1477 | | |
| QSstraight | 49 | 12.3 | 71 | 1322 | | |
| Autoclave | 64 | 0.6 | 111 | 1923 | | |

The quickstep cure cycle has demonstrated a 23–50% reduction in total processing time compared to the autoclave cure cycle without compromising composite quality [91]. Quickstep composites have also demonstrated similar mechanical properties to autoclave composites when both specimens are exposed to a damaging aging environment [92–94]. Table 5 compares the mechanical properties of quickstep samples to other composite manufacturing processes.

### 7.1. Melding

The melding process produces seamless joints with the use of quickstep technology [95]. The process represented in Figure 12 is a two-stage melding process. In the first stage, hot fluid is circulated to the region that needs to be fully cured. At the same time, cold fluid is circulated to another area of the region that is intentionally left uncured. In this way, the cold fluid prevents cross-linking of the resin, leaving parts of that material uncured. In the second stage, the uncured regions of two similarly treated materials are joined in the desired arrangement and then fully cured by applying the cure cycle. The melding process can overcome the limitations experienced with adhesive bonding and is a promising technique for repairing parts in the aerospace industry [96]. However, the mechanical integrity of the joint needs to be verified. Corbett et al. [97] evaluated the mechanical integrity of the initial transition zone after curing the complete part. Composites joined by the melded process were compared to co-cured and adhesively bonded samples. HexPly914C prepreg, impregnated with a 914C matrix, was used in the study. There was no significant variation in the fully cured areas in the melded samples; hence the final strength was not affected. The melded and co-cured samples had similar lap shear strength values of about 15 ± 0.7 MPa and 15.2 ± 0.6 MPa. The adhesively joined sample produced the largest lap shear strength of 17.8 ± 2.0 MPa. Although the initial transition zone did not affect the strength of the fully cured material, the degree of cure between the transition zone must be optimized for better bonding [98]. While partial vitrification above 105 °C before joining does not affect the final strength, SBS strength decreases rapidly at a temperature beyond 120 °C because of the higher degree of cure. Increased network formation results from a higher degree of cure, thereby reducing the joining/melding capacity of the laminates. Melded joins are optimized by maintaining material temperatures between 70 °C and 120 °C.

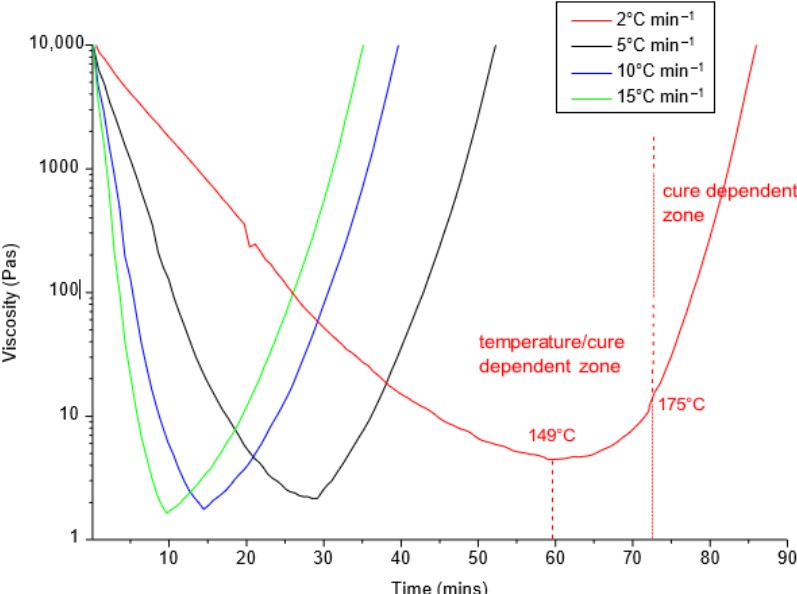

**Figure 11.** Resin viscosity as a function heating rates [99].

**Table 5.** Mechanical Properties of Composites Manufactured Using Different Processes.

| Composite System | Manufacturing Process | Fiber Vol Fraction | Void Content | ILSS | Flexural Strength | Ref. |
|---|---|---|---|---|---|---|
| Hexply 6376 prepreg | Autoclave | 64.1 | 0.6 | 111 | 1923 | [88] |
| | Quickstep | 60.2 | 1.8 | 115 | 1755 | |
| | Hot press | | | | | |
| Hexply 6376 prepreg | Autoclave | 64.1 | <1 | 111 | 1923 | [99] |
| | Quickstep | 62.3 | <1 | 121 | 1755 | |
| | Hot press | | | | | |
| Carbon fiber/epoxy 914/40/G703 | Autoclave | | | 75 | | [84] |
| | Quickstep | | | 83 | | |
| | Hot press | | | | | |
| Hexply 914 carbon fiber/epoxy | Autoclave | 54.86 | | | | [100] |
| | Quickstep | 52.17 | | | | |
| | Hot press | | | | | |
| Toughened epoxy resin MTM 45 | Autoclave | | | | | [101] |
| | Quickstep | | | 77.3 | | |
| | Hot press | | | 71.3 | | |
| Cytec cyom 977-2A prepreg | Autoclave | | | | | [87] |
| | Quickstep | 60 | 1.70 | 84 | 1258 | |
| | Hot press | 61.4 | 1.04 | 82 | 1332 | |
| Hexply M18/1carbon fiber epoxy | Autoclave | | | 64.2 | | [102] |
| | Quickstep | | | 72.2 | | |
| | Hot press | | | | | |

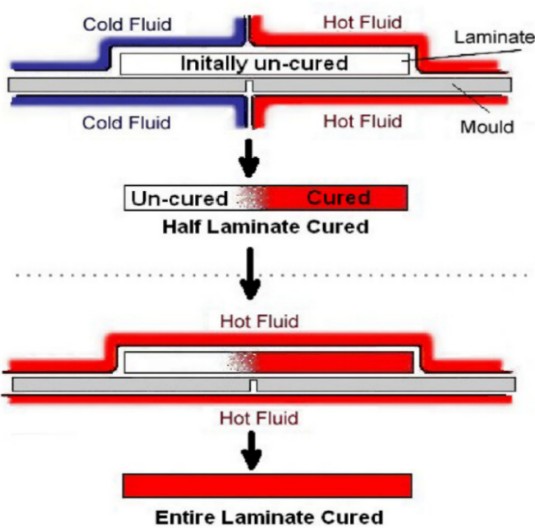

**Figure 12.** Schematics of the Melding Process [97]. Reprinted with permission from Elsevier.

*7.2. Other Quickstep Applications*

Quickstep processing has been used for various applications. Campell et al. [103] demonstrated the one-step manufacturing of a sandwich blade using the quickstep process. The quickstep cured sandwich vinyl ester prepreg had better flexural properties with fewer variations than the room temperature cured epoxy sandwich specimen. The lack of variation indicates consistent curing with the quickstep method. Furthermore, the quickstep cured sandwich sample did not show skin-core delamination between the fiber and matrix at failure, indicating improved bonding. Mujib et al. [104] investigated the possibility of manufacturing fiber-metal laminates using the quickstep (QS) manufacturing system. Polypropylene fiber/polypropylene matrix prepregs were used. The QS samples were observed to have better interfacial fracture toughness, strain to failure, and degree of adhesion properties than the hot press fabricated samples. However, samples fabricated by both methods exhibited similar tensile strength. Ogale et al. [105] reported using the quickstep process for manufacturing an Audi A1 carbon fiber composite rooftop. The developed process uses RST-resin spray transfer lay-up with a low-pressure QS curing to achieve a high surface finish at a reduced cost. Brighton et al. [106] manufactured carbon/epoxy and glass/polypropylene tubes using the quickstep process and tested the mechanical properties at different impacting speeds. The carbon/epoxy specimens showed a higher specific energy absorption of 86 kJ/Kg than the glass/polypropylene of 29 kJ/kg. Ogale et al. [107] used quickstep curing to accelerate and improve the quality of the resin infusion manufacturing process. Before resin infusion, the preform was rapidly heated and cooled after curing the composite. Rapid heating and cooling, combined with the resin infusion process, demonstrated a reduction in the manufacturing time than the autoclave and oven manufacturing processes.

## 8. Seeman Composite Resin Infusion Molding Process (SCRIMP)

The SCRIMP process is a modification of the VARTM process. It is an improved version of the VARTM process to efficiently and effectively distribute resin during impregnation using a distribution media (DM). Therefore, it is used for making high-quality and repeatable parts with minimal volatile emissions. Composite parts made by the SCRIMP process have a high fiber volume fraction typical of about 60–75% [108]. The DM is a highly permeable material placed between the vacuum bag and the topmost layer of the fabric [109]. It helps to distribute the resin quickly, thereby reducing the fill time. The resin initially flows through the DM layer before wetting the reinforcements through the thickness direction. A second type of resin distribution system exists in SCRIMP [110]. The groove-based SCRIMP incorporates channels in either the core or mold, and the resin is first delivered to the channels and then to the fiber mat. Ni et al. [111] reported the mold filling of SCRIMP based on the groove to be much faster than SCRIMP based on a highly permeable medium. Figure 13 illustrates the permeable material-based SCRIMP process. This process is used to make lightweight truck components, heavy-duty buses, large yachts, bridges [112], and naval vessels.

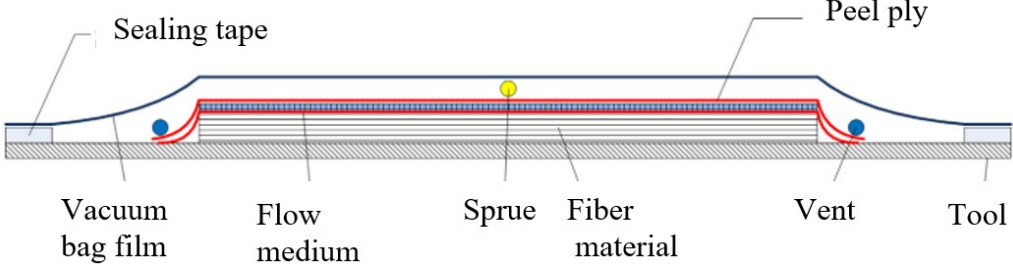

**Figure 13.** Schematics of the SCRIMP process [109]. Reprinted with permission from Elsevier.

Because SCRIMP involves the fabrication of large parts, it will be beneficial to undertake an in-depth study to optimize the SCRIMP process parameters. Optimization of the SCRIMP process parameters will minimize material waste and increase productivity. Consequently, Huan et al. [113] studied the moldability of unsaturated polyester and vinyl ester resin in SCRIMP. The rheological and kinetic behaviors of both resins were studied and compared. Results reveal that the polyester resin had longer induction and higher final conversion than the vinyl ester resin. The study proposed a model that accurately predicted the effect of inhibitors, temperature, and retarder content on gel time. The feasibility of the SCRIMP process with different fiber types and architecture has been studied. Kuraishi et al. [114] produced composite panels using the SCRIMP process with carbon fiber, glass fiber, and a carbon and glass fiber hybrid. The polyester resin was used for the three types of fiber systems. The panels manufactured by the SCRIMP process are similar to panels made by other low-cost manufacturing processes. Stiffness improved in the carbon/polyester composite compared to the glass/polyester composites. Residual stress was reported as a cause of natural resin shrinkage and thermal expansion during cure. The study demonstrated numerical and theoretical modeling for the hybrid carbon/glass polyester composite system. A carbon FRP-wood hybrid structure manufactured by SCRIMP has been reported by Pirvu et al. [115]. The structure exhibited minimum void content, a high fiber-to-resin ratio, and a strong interfacial bond with a wood substrate. Shih et al. [116] produced an aerospace-graded composite using a tackifier-assisted SCRIMP method. A commercial PT 500 tackifier was used to produce composites in the SCRIMP and RTM processes. The study shows that an increase in tackifier quantity increased the void content, thus decreasing mechanical properties. Improved mechanical properties were achieved when tackifiers were applied outside the fiber tows rather than inside. However, samples produced by both processes were found to be similar.

A void formation may occur during the manufacturing of complex geometries along the leading edge of the flow front during the SCRIMP process. Therefore, models based on Darcy's law have been developed to ensure complete impregnation of resin and avoid dry spots in the SCRIMP process [117]. Hatice et al. [117] introduced a method that uses a predictive tool to design an optimal distribution media pattern that accounts for flow variation along the edges of the insert during race-tracking. The authors used liquid injection molding simulation (LIMS) [118] first to investigate possible race-tracking (RT) scenarios. Then a depth-first search (DFS) algorithm was used to optimize the distribution media layout to minimize the worst filling RT scenarios to an acceptable tolerance. The optimal solution is experimentally validated with the experimental fill time of 221 s and a numerical fill time between 183–273 s. Because the modeling of groove-based SCRIMP is scarred, Han et al. [119] proposed a hybrid two and half and three dimensional model to simulate the resin flow in SCRIMP. The resin flowed in two regions, i.e., the fiber-free region (grooves) and the bulk fiber material region, considered in this model. The authors reported the flow patterns and filling time predicted in the simulation to coincide with experimental results. As a result, a hull was fabricated based on the optimized "fiber chops network" type channel.

### 9. Resin Film Infusion (RFI)

In the RFI process, one male or one female mold of the desired shape is used [120]. A thin film of neat resin is interleaved with layers of fibers and placed in the mold. The lay-up assembly is vacuum bagged, and the air is removed with a vacuum pump [121]. The lay-up assembly is then placed inside an oven or autoclave for curing. When the mold is heated and pressurized, the resin melts, flows into the fibers, and is then cured. The cure cycle is carefully selected to achieve the proper time-temperature-viscosity profile to ensure proper fiber saturation [122]. A schematic representation of the RFI manufacturing setup is presented in Figure 14 [123].

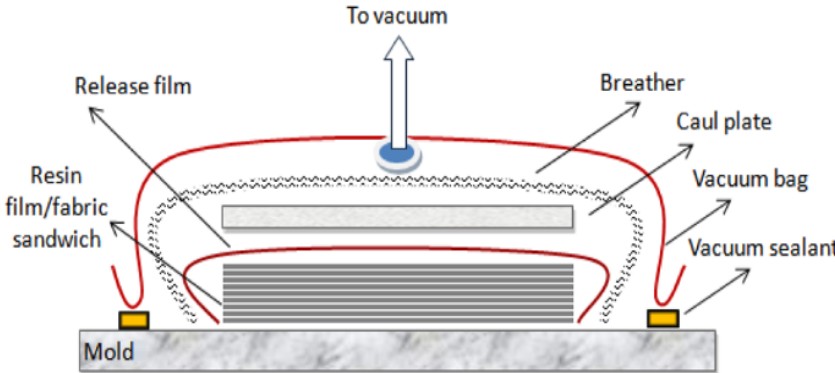

**Figure 14.** Schematic of Resin Film Infusion (RFI). Reprinted with permission from [123]. Copyright (2022) American Chemical Society.

The fiber characteristics, heating rate, resin characteristics, and cure kinetics are critical aspects that must be considered when using this method [124]. The thickness of the neat resin film depends on the permeability of fiber and the fiber volume fraction to be achieved in the composite. The RFI process manufactures composites that reach high fiber volume fraction [125]. Some of the advantages of this process are that: tooling is cheap [125], layers do not experience any form of crimp, the debulking operation is not continuously repeated when manufacturing thick composites, the resin is easily toughened, it produces high-quality composites, and dry fibers need no special storage. The drawbacks of the RFI process are that material costs are high, complex shape parts are difficult to manufacture, and the operation of RFI requires highly skilled technicians. The RFI method is used for fabricating large monolithic or sandwich structures such as stiffened skins and rib-type structures, "front nose," and "rear fender" components [126] in the automotive industry.

The resin flow and cure cycle are critical aspects of the RFI process. The resin flows across the preform thickness in the RFI process when the temperature increases and vacuum pressure is applied. Preform permeability is lower in the thickness direction than along the in-plane direction. Hence impregnating the resin into the preform is difficult in the RFI process. In manufacturing a high-quality composite using the RFI method, it is important to understand the resin flow and predict the flow behavior during cure. Darcy's law shows that increasing the pressure increases the rate at which resin infiltrates the preform and, at the same time, increases the degree to which the fiber bed is compacted. The increased compaction reduces permeability, and the rate at which resin impregnates the preform is reduced [127]. Fiber lay-up sequences have been shown to influence the flow of resin. In a unidirectional ply arrangement, the fiber tows are closely stacked, leaving no space for flow channels, unlike the cross-ply with larger flow channels that aid permeability of the resin [127]. The cure cycle and type of resin used in manufacturing the composite are also affected by resin flow. Qi et al. [128] evaluated the flow performance of four different resin systems using a dynamic viscosity profile measurement with a T-panel flow test. Darcy's law was used to predict flow and calculate a flow factor. Two cure cycles were used: the standard cure cycle with a heating rate of 2 °C/min and a 2 h cure at 177 °C with a pressure of 630 KPa and the dwell cure cycle with a modification of 30 min dwell at 140 °C. The authors observed that an increased heating rate decreases the minimum viscosity of the resin. However, a decrease in the minimum viscosity did not necessarily increase the flow factor. A faster heating rate reduces the resin's gel and flow times. The prolonged processing window resulting from the dwell time did not significantly affect the flow factor. The authors explained that this might be due to increased resin viscosity during the dwell period. Garschke et al. [129] had a different view based on their study results. Using the quickstep rapid heating technique, the authors maximized resin flow

and improved preform impregnation of a thermoplastic-toughened resin film with different curing cycles. The authors reported a slight increase in the flow factor despite gelation time being shortened due to the increased heating rate. This increase was attributed to the lower viscosity of the resin, driving the flow behavior.

Lower viscosity also plays a significant role in the flow performance to increase the flow factor during hold time. Nevertheless, the viscosity-lowering effect appears to be insignificant beyond 8 °C/min, as demonstrated by Garschke et al. [130] in another study when they predicted the viscosity-time-temperature behavior of resin film using chemo-rheological modeling. The quantity of resin infiltrating and bleeding from the preform must be controlled. Increased resin quantity for a constant cure condition and bleed-out hole ratio decreases fiber volume fraction. On the other hand, an increase in bleed-out hole ratio for a constant resin quantity resulted in a high fiber volume fraction. However, the high bleed-out hole ratio leads to a significant increase in resin starvation and porosity. Improved surface quality composites are produced when the bleed-out hole ratio is optimally reduced, and a vacuum is applied during curing for a more extended period [131].

The RFI process has been extensively applied in the co-curing approach. The co-curing process integrates two molding processes: the prepreg/autoclave process and the liquid resin infusion (co-LRI) or resin transfer molding (co-RTM) process. The co-curing process is a cost-efficient fabricating method that manufactures composite structures of high performance. The co-curing process manufactures large and complex structures in one cure cycle, thus reducing equipment and energy consumption. The process also reduces the number of joints in a structure, increasing surface smoothness and mechanical properties. The procedures involved in co-curing are few, resulting in time and labor-saving. The co-RFI process combines the prepreg/autoclave and RFI processes. The RFI and prepreg resins' compatibility and interface are critical to fabricating a quality composite. Xuqiang et al. [132] used the co-RFI process to manufacture a stiffener and prepreg skin structure. Four types of carbon fiber prepreg with one epoxy film were used. Results indicate that the co-RFI process can achieve improved mechanical property at the co-curing interface region in composite manufacture with good processing conditions. Therefore, the authors suggested that the process could be used for fabricating high flexible composite structures. In another study, Xuqiang et al. [133] confirmed that co-RFI laminates have improved mechanical properties over either the RFI or prepreg laminates. They reported no weakening effect in the interface region. Factors such as resin film infusion part, lay-up type of the prepreg part, epoxy tackifier of fiber preform, and isothermal dwell have been demonstrated to affect the quality of laminate produced by the co-RFI process [134].

Voids and resin-rich regions occur most in the RFI part and the interlaminar interface between the RFI and prepreg parts. The void and resin-rich zone in the co-cured interface reduces the delamination resistance. The critical interlaminar fracture toughness exists between RFI laminates and the prepreg laminates. The $G_{IC}$ at the crack propagation stage is higher due to crack deflection and fiber bridging in the co-RFI laminate. Void occurrence can be minimized by laying up the prepreg part below the RFI part. The dwell time provides prolonged processing at low resin viscosity for the resin's impregnation of the fibers. Epoxy tackifier increases the resin-rich region at the co-cured interface, affecting the interlaminar fracture toughness.

The RFI process is reported as an effective and cost-efficient method to infuse CNT-modified resin into a fiber-reinforced composite. Proper CNT infusion is possible because the RFI process can alter the rheological characteristics of the matrix polymer. The RFI process is used to investigate the infiltration of CNTs into the fiber tows using two lay-up strategies, i.e., the grouped and interleaved lay-up [135]. Composite fabricated by the interleaved lay-up had CNTs more uniformly distributed, higher fiber volume fraction, and better compressive and electrical properties. However, the interleaved composite exhibited higher void content due to the lack of air removal. Void content can be minimized by creating channels for air removal. The effective alignment of CNTs along the through-thickness direction has been demonstrated by a multi-layer RFI (MLRFI) method [136].

The MLRFI uses the resin flow to align the nano-filler along the through-thickness direction during fabrication. The preferential alignment led to CNTs stitches between the fiber tows responsible for the improved fracture toughness observed in composites fabricated by the MLFRI method. The improved fracture toughness is attributed to a higher $G_{IC}$, crack propagating along the CNTs/matrix interface leading to more energy dissipation and a combination of intralaminar and interlaminar crack propagation due to CNTs perpendicular stitching across fibers (Figure 15). The red region in Figure 15 illustrates the CNTs stitching between layers of fiber tows (grey cylinders) embedded in a resin matrix (grey rectangle), which results in hybrid crack propagation. RFI has been used to fabricate nanosilica reinforced epoxy-based hybrid composites [137]. The nanosilica-filled hybrid CFRP and GFRP exhibited a 35% and 30% increase in compressive strength, respectively. The authors reported low void content in nanocomposites fabricated by the RFI method. A percolating-assisted resin film infusion method is used to fabricate a nanocomposite material that can be used for various surface protection applications such as flame retardation, deicing, and microwave irradiation [138]. The method uses a sealant tape to limit the flow of reduced graphene oxide (RGO) flow into the fibrous preform, thus accumulating RGO on the surface of the CFRP surface. The enriched RGO deposited on the CRFP surface allows high conductivity, reducing lighting damage on the surface.

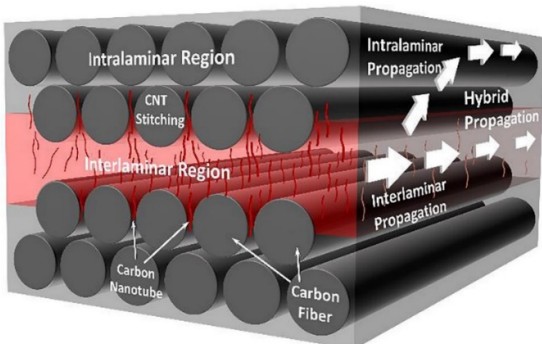

**Figure 15.** Illustration of crack propagation route in RFI fabricated nanocomposite [136]. Reprinted with permission from Elsevier.

## 10. Resin Infusion under Double Flexible Tooling (RIDFT)

The RIDFT process is a variation of the liquid composite molding (LCM) technique. The RIDFT idea was developed to solve problems existing in other LCM processes. Some of these issues are high tooling costs, slow production rates, complex resin infusion, long processing times, usage of an expensive preform, and environmental pollution [139]. The RIDFT process uses a two-dimensional resin flow to produce cost-effective composite parts at an increasingly higher production rate while reducing volatile organic compound emissions into the environment. Figure 16 illustrates the different stages that comprise the RIDFT process. Fiber reinforcements are initially placed between the two silicone diaphragms and closed (step 1). Air is vacuumed from between the two silicone sheets via the vent port to compact the fiber reinforcement, and therefore, permeability is reduced (step 2). Once the resin infusion gate is open, vacuum pressure drives the resin from the reservoir to impregnate the fiber reinforcement (step 3). A flow distribution media is placed on top of the silicone sheets to increase permeability and assist in the quick infiltration of the resin. After impregnation, the infusion gate is closed, and the wetted reinforcement inside the silicone sheets is draped over a one-sided mold with the aid of a vacuum [140] (step 4). At this time, the vent port is still left open. The formed part is allowed to cure, after which it is de-molded (step 5). Using a silicone sheet prevents the direct contact of the wetted reinforcements on the mold, which increases the tool life [141]. However, silicone sheets are expensive to replace, and cleaning them during production

runs between parts takes longer. The various stages of the RIDFT process are shown in Figure 15.

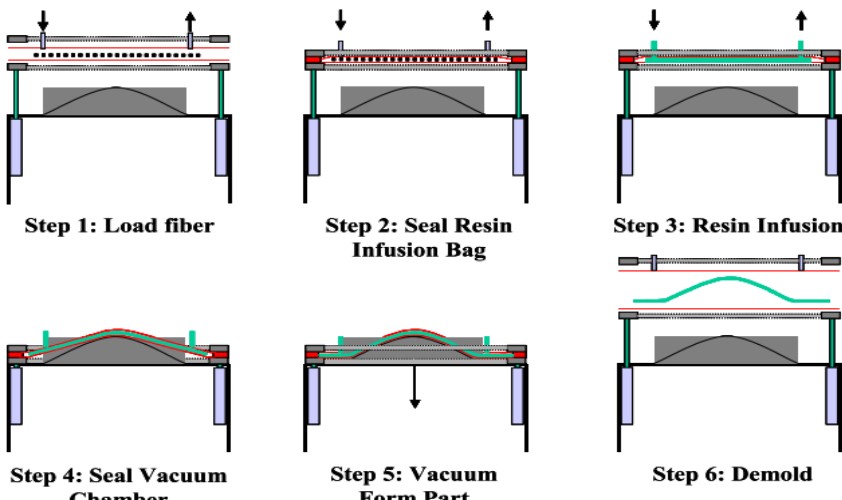

**Figure 16.** Schematic of the RIDFT Process [139]. Reprinted with permission from Elsevier.

The automobile industries seek efficient, cost-competitive, and environmentally friendly composite manufacturing processes, as in the case of the RIDFT, to reduce the high cost of composite painting. The in-mold coating (IMC) technique presents an alternative to conventional painting methods. The IMC is a process whereby manufactured parts are decorated during the molding cycle. Chiu et al. [142] proposed an in-mold coating method for the RIDFT process. The method infused polyurethane paint and vinyl ester resin liquid simultaneously into the flexible tooling. The polyurethane paint is meant to coat the composite, while the vinyl ester serves as the matrix. A separation layer prevents the two liquids from mixing during infusion and curing. Composite parts manufactured by the proposed RIDFT IMC method were compared with parts manufactured with the RIDFT and manually painted. Results indicate that 45% and 55% were saved in capital investment and time, respectively. Rather than using liquid paint, Toro et al. [143] investigated the use of a paint film for implementing IMC in the RIDFT process. The study aimed to establish process parameters for the adequate adhesion of paint films to composite parts during the forming process. The authors reported that the heating duration and temperature affected the paint quality. The temperature necessary to soften the paint film cured the resin prematurely and affected the formability of the fiber-resin assembly. Despite the lack of the paint film's adhesion to the parts, the inclusion of the paint film was observed not to have affected the resin flow. Therefore, the authors suggested using paint films with better adhesion characteristics. To advance their research, Carlos et al. [144] alternatively assessed the use of thermo-formable polycarbonate paint film for the RIDFT IMC. The study evaluated the forming capability and surface quality finish of fabricated composite parts. Increasing the temperature yielded better drapability; however, print-through of the fibers through the film was observed.

Process parameters such as temperature, mold type, time, fiber reinforcement, and vacuum pull were optimized, with print-through being the response variable. Results revealed minimal print-through at a forming temperature of 147 °C, and the film formability had limited subtle contours. On the other hand, at 160 °C, the forming capability of the composite assembly was maximized, while the surface finishes had high print-through.

The Carlos et al. [144] study suggests that successful implementation of IMC techniques with the RIDFT process may be attributed to the heating temperature. The RIDFT machine uses a UV lamp to cure composite parts. The goal of using UV-curing is to reduce the production cycle time by shortening the cure cycle time. Studies have been carried out

to determine the efficiency of UV curing of 2D and 3D composite structures manufactured by the RIDFT. For instance, Augustine et al. [145] evaluated the feasibility of designing and incorporating a cure on demand (COD) system in place of the organic peroxide-based catalyst curing system for the RIDFT process. The study was undertaken to optimize the production cycle time. UV-laminates were produced at a fraction of the time required to produce catalyst-cured laminates. Results indicated that the UV laminates were comparable and, in some instances, had better mechanical properties than those produced by the catalyst curing system. However, the authors used a flat 2D component in conducting the experiments. To validate their findings, Adewuyi et al. [146] investigated the feasibility of UV curing of a 3D curvilinear component in the RIDFT process. The study aimed to identify the optimum UV-lamp position for curing 3D composite components with minimal UV intensities. Three different manufacturing designs and four predicted lamp positions were used to determine the optimal position with the best mechanical properties. Although the optimal lamp position demonstrated more improved mechanical properties than the laminate cure with a catalyst, uniform curing over the entire component geometry could not be achieved. As a result, further work was proposed in this area.

The implementation of IMC and incorporating the UV-curing technique in the RIDFT process results in increased efficiency of the overall system. However, the quality of composites produced with the RIDFT process still needs improvement. Parker et al. [147] manufactured a nanocomposite structure using the RIDFT process to address this mechanical integrity issue. Numerous studies have demonstrated the use of nanomaterials to improve the mechanical properties of composite structures. The Parker et al. [147] study showed that the addition of 2 wt.% carbon nanotube (CNT) increases the mechanical property of composites manufactured with the RIDFT process. The study also investigated the optimum flow distribution channel (FDC) pattern that can infuse CNT-rich resin of 10,000 cPa viscosity. Similarly, Divyesh et al. [148] reported the manufacturing of CNT reinforced glass fiber plastic (CNT_GFRP) using the RIDFT process. The study demonstrated the use of stretchlon film with the RIDFT process to reduce CNT_GFRP production cycle time by 42% and increase UTS by 12%. However, the study reported a decrease in flexural strength with CNT_GFRP samples produced with stretchlon film.

Table 6 compares the out-of-autoclave manufacturing processes discussed in this review. Their principle of operation, advantages, and disadvantages are highlighted.

**Table 6.** OoA manufacturing comparison.

| | Principle | Advantage | Disadvantage |
|---|---|---|---|
| **VBO** | Partial impregnated pre-preg is cured in an oven | Low investment<br>Large number of parts per production cycle<br>Easy to use<br>Similar part quality with autoclave | Long processing time<br>Lesser heat transfer than autoclave<br>Comparatively high energy consumption<br>Limited space, hence batch production. |
| **RTM** | In plane resin infusion (with injection pressure) using two sided-mold | Lesser void content<br>Produce complex parts<br>Two side has high surface finished | Labor intensive preform preparation<br>High tooling cost<br>3D resin flow problematic to control<br>High processing cycle time |
| **VARTM** | In plane resin infusion (with vacuum pressure) using two sided-mold | Cheap raw material<br>Low tooling cost<br>2D impregnation easy to control<br>Reduce volatile emission | High process complexity<br>Limitation to achieve desirable fiber volume fraction<br>Require low viscosity resin<br>One sided high surface finish |
| **SCRIMP** | In plane resin infusion (with vacuum pressure) using two sided-mold, with flow media | Higher dimensional consistency<br>Better quality products than RTM<br>Reduce volatile emission<br>Use for large scale structure | Propriety process<br>One side finished<br>High permeability preforms required |
| **Quickstep** | Vacuum bagged laminate is cured using high heating from heat transfer fluid (HTF) | Higher heat transfer rates<br>Use Low pressure<br>Precise temperature control<br>Restraint of exotherm | Uses HTF for Heat transfer<br>Limited to medium complexity parts<br>Limited life span of flexible membrane<br>Expensive material |
| **RFI** | Through-thickness resin infusion, with resin film between fiber layers | Produce good quality parts<br>Resin easily toughened<br>Repeated debulking operation not need | Hard to fabricate complex shaped parts<br>Resin film placement inside mold requires high labor |
| **RIDFT** | In plane resin infusion (with vacuum) using two sided-flexible molds | Uses low-cost resin materials<br>Easily control 2D impregnation | Flexible tooling may be easily damaged<br>Requires low viscosity resin |

## 11. Future Research Direction

The recently proposed pressurized infusion method for a heated-VARTM process applies an external pressure to increase the compaction of fiber preform while removing excess resin. The study demonstrates that resin flushing should be applied to improve part quality and mobilize and remove process-induced voids after the mold is completely filled. Results revealed a 0.02 void content present in the 6-ply sample laminate. However, the authors reported that the 12- and 18-ply laminates could not be characterized due to the characterization technique used. The flatbed scanner used in the study was sensitive to thickness. Transparency critically decreased as laminate thickness increased. The microstructural assessment of a thicker laminate will confirm the credibility of the proposed pressurized infusion method. Furthermore, the concept was investigated with a flat 2D laminate sample reported in the study. Applying this same idea to manufacture a three-dimensional structure could possibly lead to a different outcome in mechanical properties, considering the effect of corners on mold filling and residual stress. In addition, the report stated that the idea had been successfully applied to small and medium-sized 2D structures; applying it to large parts can magnify errors and yield different results. Finally,

it is worth reducing the resin flush duration to see if the same mechanical properties can be achieved to increase efficiency and productivity. Further research is recommended to provide information that will answer these questions; to provide credibility and acceptance to applying external pressure and resin flushing in a heat-VARTM process. If this method is credible, then applying it to the RIDFT process should be investigated. Applying compaction pressure to improve the mechanical properties of fabricated laminate using the RIDFT process should also be studied. In addition, an improved cure method should be developed for the RIDFT to enhance the even distribution of three-dimensional parts fabricated with the RIDFT.

## 12. Conclusions

In this review, we have summarized and presented the literature relating to common out-of-autoclave processes for the manufacturing of composite structures. Composite materials are a combination of two distinct materials to form one superior material that is better than its constituent material alone. This material is unique for its high specific strength and stiffness properties compared to conventional aluminum and steel. Therefore, they find application in diverse industries such as aerospace, automotive, renewable energy, and marine, to mention a few. Composite is a combination of fiber and matrix. The fibers serve as load carriers while the matrix resin distributes the load to the fiber while protecting it from weather conditions. The use of prepreg is a more advanced way of making composite materials. Prepregs are reinforced fibers with stage B resins that can easily be maneuvered into open molds to make complex geometries. Prepregs can be manually or automatically laid. The vacuum bagging system mainly uses prepregs, and components are cured in the autoclave to improve mechanical properties. Autoclave curing simultaneously applies vacuum, pressure, and temperature to consolidate the stacked prepregs.

Autoclave processing has been used for decades because of the high performance and quality composites it produces. The VBO oven cure process uses a partially impregnated prepreg to produce a composite similar to that of the autoclave. New generation resins have been developed to make composite production effective and efficient and increase the mechanical strength of parts made from VBO prepreg. Some out-of-autoclave processing includes RTM, VARTM, Quickstep, SCRIMP, RFI, and RIDFT. The RTM is a closed-mold system that uses positive pressure to drive liquid resin into the preform between the closed mold. On the contrary, VARTM, with a one-sided molding system, uses vacuum pressure to impregnate the fiber preform with resin. The quickstep cure method applies a high heating rate to reduce resin viscosity in composite manufacturing for improved mechanical properties. The SCRIMP process uses a distribution media to efficiently wet and advance the resin's flow front, minimizing flow variation. Active and passive control algorithms have been used to optimize distribution media layout. The RFI technique uses a resin film placed between fibers to manufacture parts. On the contrary, the RIDFT process uses vacuum pressure to impregnate reinforcements embedded between two flexible diaphragms with a 2D flow. The RIDFT process is reported to improve productivity and profit. The above OoA processes have demonstrated cost-effectiveness, reduced environmental impacts, and less energy consumption compared to the autoclave method. However, additional research and development are required to improve the quality of composite fabricated with the OoA process.

**Author Contributions:** Conceptualization, O.A.E.; methodology, O.A.E.; validation, O.A.E.; formal analysis, O.A.E.; investigation, O.A.E.; resources, O.A.E.; writing-original draft preparation, O.A.E., N.A., V.O.E. and O.I.O.; writing-review and editing, V.O.E. and O.I.O.; visualization, V.O.E. and O.I.O.; supervision, V.O.E. and O.I.O.; project administration, O.I.O.; funding acquisition. All authors have read and agreed to the published version of the manuscript.

**Funding:** This research was funded by the National Science Foundation, Grant Number# 1950500.

**Informed Consent Statement:** Not applicable.

**Data Availability Statement:** Not applicable.

**Conflicts of Interest:** The authors declare no conflict of interest.

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
