# Peer review of "A Review on the Out-of-Autoclave Process for Composite Manufacturing"

_jcs, doi:10.3390/jcs6060172_

Round 1

Reviewer 1 Report

The paper in question is a very extensive review about Out-of-Autoclave manufacturing techniques. In the opinion of the reviewer, the authors have partially facilitated the writing of the review by just citing other authors. Thus, some information are redundantly or at least in a very similar way published. It should be the other way around that statements and facts are listed once with several references.

The references are not contemporary! Most of the references are 5 years and older. 

A final table to compare the OoA-manufacturing techniques would be advantageous for the reader. It should be pointed out, that e.g. RTM-parts have two high quality surfaces and e.g. VARTM-parts have only one.

Author Response

Point 1: The paper in question is a very extensive review about Out-of-Autoclave manufacturing techniques. In the opinion of the reviewer, the authors have partially facilitated the writing of the review by just citing other authors. Thus, some information are redundantly or at least in a very similar way published. It should be the other way around those statements and facts are listed once with several references

Response 1: Similar findings from articles previously discussed extensively have been summarized in one idea, supported by the facts and then followed by the references. One example is in the Quickstep section i.e., chapter 7

Point 2: The references are not contemporary! Most of the references are 5 years and older

Response 2: Over ten recently published Journal articles have been reviewed, discussed and included in the manuscript

Point 3: A final table to compare the OoA-manufacturing techniques would be advantageous for the reader. It should be pointed out, that e.g., RTM-parts have two high quality surfaces and e.g., VARTM-parts have only one

Response 3: A final table making the comparison between the OoA-manufacturing techniques is presented at the end of the manuscript before chapter 11. The table highlighted the core principle, advantages and disadvantages of each OoA process discussed in the manuscript. Furthermore, the RTM-parts are indicated to have two-sided high-quality finish as an advantage, while the VARTM-parts has one-sided high-quality finish as a disadvantage.

Comments PDF response is attached.

Reviewer 2 Report

This paper makes a short review on the OOA process. In each section, each individual process was discussed. It is a good reviewing paper. Some comments are listed below.

  1. The numbering of the figures is out of order.
  2. The VBO process using OOA prepreg attracted many researchers recently, and many papers are published. The authors are suggested to discuss more issues in this section.
  3. One new type of resin infusion process was published in “Carbon/Epoxy Composites Fabricated by Vacuum Consolidation of the Interleaved Layup of Prepregs and Dry Fibers, Fibers and Polymers, 2021, 22(2),” which may need discussion.

Author Response

Point 1: The numbering of the figures is out of order

Response 1: The Figure numbering has been properly ordered

Point 2: The VBO process using OOA prepreg attracted many researchers recently, and many papers are published. The authors are suggested to discuss more issues in this section

Response 2: Additional ten recently published Journal articles on VBO process using OOA prepreg has been reviewed, discussed and included in the manuscript

Point 3: One new type of resin infusion process was published in “Carbon/Epoxy Composites Fabricated by Vacuum Consolidation of the Interleaved Layup of Prepregs and Dry Fibers, Fibers and Polymers, 2021, 22(2),” which may need discussion

Response 3: The Journal article titled “Carbon/Epoxy Composites Fabricated by Vacuum Consolidation of the Interleaved Layup of Prepregs and Dry Fibers, Fibers and Polymers, 2021, 22(2),” has been discussed and included in the manuscript

Round 2

Reviewer 1 Report

I found five comments which had been addressed:

1. The resin types are not introduced before the experiments

2. The bleeder material has to be depicted in Fig. 2

3. Use always "bar" instead of "atm"

4. There is no injection pressure in VARTM simply because there is no injection

5. Fig. 15: It needs some explanation. E.g., what is the grey rectangle!

Author Response

Dear Ms. Emily,

Thank you for your email dated May 10, 2022. We are happy to resubmit the revised version of Manuscript ID: JCS-1732139 entitled" A Review on the Out-of-Autoclave Process for Composite Manufacturing" in consideration for publication in the Journal of composite science.

The authors thank the reviewers for the comments, which we believe will increase the clarity of the manuscript for the reader. The points have been addressed, and changes have been made in the accompanying manuscript. Additionally, the authors would like to directly address or comment on each point made by the reviewer below. For clarity, our response is in colored lettering.

Round 2 Revision

Point 1. The resin types are not introduced before the experiments

Response 1:  The fiber/resin and prepreg system for the various experimental work cited in the manuscript has been introduced prior to the stated findings

Point 2. The bleeder material has to be depicted in Fig. 2

Response 2: The bleeder material depicted in fig. 2 has been indicated, in a bracket i.e., woven polyester fabric.

Point 3. Use always "bar" instead of "atm"

Response 3: “bar” has been replaced with “atm” for example in figure 3

Point 4. There is no injection pressure in VARTM simply because there is no injection

Response 4: That was the inlet pressure and it has been corrected

Point 5. Fig. 15: It needs some explanation. E.g., what is the grey rectangle!

Response 5:  A better explanation is provided for readers clarity: “The red region in Figure 15 illustrates the CNTs stitching between layers of fiber tows (grey cylinders) embedded in a resin (grey rectangle), which results in hybrid crack propagation”

Reviewer 2 Report

I have no further questions.